# Communication-Efficient Distributed Learning of Discrete Probability Distributions

**Ilias Diakonikolas**
CS, USC
diakonik@usc.edu

**Elena Grigorescu**
CS, Purdue
elena-g@purdue.edu

**Jerry Li**
EECS & CSAIL, MIT
jerryzli@mit.edu

**Abhiram Natarajan**
CS, Purdue
nataraj2@purdue.edu

**Krzysztof Onak**
IBM Research, NY
konak@us.ibm.com

**Ludwig Schmidt**
EECS & CSAIL, MIT
ludwigs@mit.edu

## Abstract

We initiate a systematic investigation of *distribution learning* (*density estimation*) when the data is *distributed* across multiple servers. The servers must communicate with a referee and the goal is to estimate the underlying distribution with as few bits of communication as possible. We focus on non-parametric density estimation of discrete distributions with respect to the $\ell_1$ and $\ell_2$ norms. We provide the first non-trivial upper and lower bounds on the communication complexity of this basic estimation task in various settings of interest. Specifically, our results include the following:

1. When the unknown discrete distribution is *unstructured* and each server has only *one* sample, we show that any *blackboard* protocol (i.e., any protocol in which servers interact arbitrarily using public messages) that learns the distribution must essentially communicate the entire sample.

2. For the case of *structured* distributions, such as $k$-histograms and monotone distributions, we design distributed learning algorithms that achieve significantly better communication guarantees than the naive ones, and obtain tight upper and lower bounds in several regimes. Our distributed learning algorithms run in near-linear time and are robust to model misspecification.

Our results provide insights on the interplay between structure and communication efficiency for a range of fundamental distribution estimation tasks.

## 1 Introduction

### 1.1 Background and Motivation

We study the problem of *distribution learning* (or *density estimation*) in a *distributed model*, where the data comes from an unknown distribution and is partitioned across multiple servers. The main goal of this work is to explore the inherent tradeoff between *sample size* and *communication* for non-parametric density estimation of discrete distributions. We seek answers to the following questions: What is the minimum amount of communication required to learn the underlying distribution of the data? Is there a communication-efficient learning algorithm that runs in polynomial time? We obtain the first non-trivial algorithms and lower bounds for distributed density estimation. Before we state our results, we provide the relevant background.

**Density Estimation.** Distribution learning or density estimation is the following prototypical inference task: Given samples drawn from an unknown target distribution that belongs to (or is

well-approximated by) a given family of distributions $\mathcal{P}$, the goal is to approximately estimate (learn) the target distribution. Estimating a distribution from samples is a fundamental unsupervised learning problem that has been studied in statistics since the late nineteenth century [36]. The classical statistics literature focuses primarily on the sample complexity of distribution learning, i.e., on the information-theoretic aspects of the problem. More recently, there has been a large body of work in computer science on this topic with an explicit focus on computational efficiency [12, 11, 7, 8, 1, 13, 2]. We emphasize that the aforementioned literature studies density estimation in the centralized setting, where all the data samples are available on a single machine.

**Distributed Computation.** In recent years, we have seen an explosion in the amount of data that has been generated and collected across various scientific and technological domains [10]. Due to the size and heterogeneity of modern datasets, there is a real need for the design of efficient algorithms that succeed in the distributed model, when the data is partitioned across multiple servers. A major bottleneck in distributed computation is the *communication* cost between individual machines. In practice, communication may be limited by bandwidth constraints and power consumption, leading to either slow or expensive systems (see, e.g., [23] for a survey). Hence, the general problem of designing communication-efficient distributed protocols is of fundamental importance in this setting. In recent years, a number of statistical estimation problems have been algorithmically studied in the distributed setting [3, 16, 15, 40, 21, 30, 24, 33, 5, 29]. To the best of our knowledge, the problem of nonparametric density estimation has not been previously studied in this context.

**This Work: Distributed Density Estimation.** We initiate a systematic investigation of density estimation in the distributed model. We believe that this is a fundamental problem that merits investigation in its own right. Also, the problem of distributed density estimation arises in various real-data applications when it is required to reconstruct the data distribution from scattered measurements. Examples include sensor networks and P2P systems (see, e.g., [35, 32, 27, 41, 37] and references therein).

We explore the tradeoff between communication and statistical efficiency for a number of fundamental nonparametric density estimation problems. Specifically, we insist that our algorithms are sample-efficient and our goal is to design distributed protocols using a minimum amount of communication. As our main contribution, we provide the first non-trivial upper and lower bounds on the communication complexity of density estimation for a range of natural distribution families that have been extensively studied in the centralized regime. The main conceptual message of our findings is the following: *When the underlying discrete distribution is* unstructured*, no non-trivial communication protocol is possible. In sharp contrast, for various families of* structured *distributions, there are non-trivial algorithms whose communication complexity significantly improves over naive protocols.* It should be noted that all our algorithms are in addition computationally efficient.

**Communication Model for Density Estimation.** We now informally describe the communication model used in this paper. We refer to the preliminaries in Section 2 for formal definitions.

The model is parameterized by the number of samples per server (player), which we denote by $s$. There are a specific number of servers, each holding $s$ independent samples from an unknown distribution $P$. We call these servers *sample-holding players*. Additionally, there is a server that holds no samples from $P$. We call this server a *referee* or *fusion center*. In communication protocols considered in this work, servers exchange messages, and at the end of the protocol, the referee outputs an accurate hypothesis distribution $\widehat{P}$. More precisely, we want the the hypothesis $\widehat{P}$ to satisfy $d(\widehat{P}, P) \leq \epsilon$ with high probability (over the samples and internal randomness), where the metric $d$ is either the $\ell_1$-norm (statistical distance) or the $\ell_2$-norm.

We study two variants of this model. In the *simultaneous* communication model, each sample-holding player sends a message (of one or more bits) to the referee *once*, based only on the samples she holds and public randomness. In the *blackboard* model, the sample-holding players' messages are public, and the communication protocol does not restrict the number of times a player may speak. The goal is to minimize the amount of communication between the players and the referee, while transmitting enough information about the samples so that the underlying distribution $P$ can be approximately recovered from the transcript of the communication.

Table 1: Communication complexity bounds for density estimation of unstructured distributions (for success probability 9/10)

| Regime | $\overline{\mathcal{CC}}_{s,1/10}(\mathrm{ADE}(\mathcal{D}_n, 1, \varepsilon, \alpha))$ | $\mathcal{CC}^{\rightarrow}_{s,1/10}(\mathrm{ADE}(\mathcal{D}_n, 1, \varepsilon, \alpha))$ |
|:---:|:---:|:---:|
| $s = 1$ | $\Omega(\frac{n}{\varepsilon^2} \log n)$ | $O(\frac{n}{\varepsilon^2} \log n)$ |
| $s = \Theta(n)$ | $\Omega(n \log \frac{1}{\varepsilon})$ | $O(\frac{n}{\varepsilon^2})$ |
| $s = \Theta(\frac{n}{\varepsilon^2})$ | $\Omega(n \log \frac{1}{\varepsilon})$ | $O(n \log \frac{1}{\varepsilon})$ |

## 1.2 Our Contributions

In this section, we provide informal statements of our main results. For the formal statements of all our results the reader is referred to the full version of the paper. We will require the following notation.' We use $n$ to denote an upper bound on the domain size of our distributions and $\alpha$ to denote the total sample size. Without loss of generality, we will assume that the domain of the distributions is the set $[n] := \{1, 2, \ldots, n\}$. The $\ell_1$ (resp. $\ell_2$) distance between two discrete distributions is the $\ell_1$ (resp. $\ell_2$) norm of the difference between their probability vectors. We note that the sample sizes in this section correspond to high constant probability of success. This can be boosted to high probability by standard techniques.

We start by pointing out the baseline result that we compare against. The naive protocol to perform distribution density estimation is the following: all the servers (players) communicate their entire sample to the referee, who applies a centralized estimator to output an accurate hypothesis. The communication complexity of this approach is $\Theta(\alpha \log n)$ bits. The obvious question is whether there exists a protocol with significantly smaller communication complexity.

**Unstructured Discrete Distributions.** Our starting point is the basic setting in which the underlying distribution over $n$ elements is potentially arbitrary and each server (player) holds exactly *one* sample from an unknown distribution over a domain of size $n$. (This basic setting is motivated by practical applications, e.g., aggregation of cell-phone data, etc.) In the centralized setting, it is a folklore fact (see, e.g., [19]) that $\Theta(n/\varepsilon^2)$ samples are necessary and sufficient to learn an unstructured distribution supported on $n$ elements within $\ell_1$-error $\varepsilon$. This fact in turn implies that the naive distributed protocol uses $O(\frac{n}{\varepsilon^2} \log n)$ bits. We show that this protocol is best possible, up to constant factors:

**Theorem 1.** *Suppose $\Theta(n/\varepsilon^2)$ samples from an unknown distribution $P$ over $[n]$ are distributed such that each player has exactly one sample. Then learning $P$ within $\ell_1$-distance $\varepsilon$ requires $\Omega((n/\varepsilon^2) \log n)$ bits of communication in the blackboard model.*

We remark that a blackboard model captures a very general interaction between sample-holding players and the referee. The players are allowed to send messages in arbitrary order and share partial information about their samples from $[n]$, perhaps using much fewer than $\log n$ bits. For instance, if one of the players has revealed her sample, other players may just notify everyone that they hold the same (or a correlated) sample, using $O(1)$ extra bits. Thus, our lower bound excludes the possibility of non-trivial protocols that do better than essentially having each machine transmit its entire sample. This statement might seem intuitively obvious, but its proof is not straightforward.

By a standard packing argument, we also show a communication lower bound of $\Omega(n \log \frac{1}{\varepsilon})$ for all protocols that estimate an unstructured discrete distribution over $[n]$ in $\ell_1$-distance. In the regime where there are $\Theta(n/\varepsilon^2)$ samples per machine, we show that there is a simple estimator that achieves this lower bound. (See Table 1 for instantiations of the theorems, and Section 2 for the formal definitions.)

**Structured Discrete Distributions.** In contrast to the unstructured case, we design non-trivial protocols that significantly improve upon the naive protocols in several regimes of interest.

Our main algorithmic results are the first communication-efficient algorithms for robust learning of histogram distributions. A $k$-*histogram* distribution over $[n]$ is a probability distribution that is piecewise constant over some set of $k$ intervals over $[n]$. Histograms have been extensively studied in statistics and computer science. In the database community, histograms constitute the most common

tool for the succinct approximation of data [9, 38, 25, 26, 1]. In statistics, many methods have been proposed to estimate histogram distributions in a variety of settings [22, 34, 17, 31].

The algorithmic difficulty in learning histograms lies in the fact that the location and "size" of these intervals is a priori unknown. In the centralized setting, sample and computationally efficient algorithms for learning histograms have been recently obtained [7, 8, 2]. Our distributed learning algorithm for the $\ell_1$-metric builds on the recent centralized algorithm of [2]. In particular, we have the following:

**Theorem 2.** *For the problem of learning $k$-histograms with $\ell_1$ error $\varepsilon$, the following hold:*

1. *In the regime of one sample per player, there exists a protocol that uses $O(\frac{k}{\varepsilon}\log n + \frac{k}{\varepsilon^3}\log\frac{k}{\varepsilon})$ bits of communication. Furthermore, any successful protocol must use $\Omega(k\log\frac{n}{k} + \frac{k}{\varepsilon^2}\log k)$ bits of communication.*

2. *In the regime of $\Theta(\frac{k}{\varepsilon^2})$ samples per player, there exists a successful protocol with $O(\frac{k}{\varepsilon}\log n)$ bits of communication. Furthermore, any protocol must use $\Omega(k\log\frac{n}{k} + k\log\frac{1}{\varepsilon})$ bits of communication.*

We now turn our attention to learning under the $\ell_2$-metric. Previous centralized algorithms for this problem [1] work in a "bottom-up" fashion. Unfortunately, this approach does not seem amenable to distributed computation for the following reason: it seems impossible to keep track of a large number of intervals with limited communication. Instead, we devise a new "top-down" algorithm that starts with a small number of large intervals and iteratively splits them based on the incurred $\ell_2$-error. A careful application of this idea in conjunction with some tools from the streaming literature—specifically, an application of the Johnson-Lindenstrauss tranform to estimate the $\ell_2^2$ error using few bits of communication—yields the following result:

**Theorem 3.** *For the problem of learning $k$-histograms with $\ell_2$ error $\varepsilon$, the following hold:*

1. *In the regime of $s = \tilde{O}(k\log n)$ samples per player, there exists a protocol that uses $O(\frac{1}{\varepsilon^2}\log n)$ bits of communication. Furthermore, any successful protocol must use $\Omega(k\log\frac{n}{k} + \frac{1}{\varepsilon}\log\varepsilon k)$ bits of communication.*

2. *In the regime of $s = \omega(k\log n)$ samples per player, there exists a protocol with $\tilde{O}(\frac{k}{s\varepsilon^2}\log n)$ bits of communication. Furthermore, any successful protocol must use $\Omega(k\log\frac{n}{k} + \frac{1}{\varepsilon}\log\varepsilon k)$ bits.*

We remark that the above algorithms are robust to model misspecification, i.e., they provide near-optimal error guarantees even if the input distribution is only *close* a histogram. As an immediate corollary, we also obtain communication efficient learners for all families of structured discrete distributions that can be well-approximated by histograms. Specifically, by using the structural approximation results of [6, 7, 20], we obtain sample-optimal distributed estimators for various well-studied classes of structured densities including monotone, unimodal, log-concave, monotone hazard rate (MHR) distributions, and others. The interested reader is referred to the aforementioned works.

For specific families of structured distributions, we may be able to do better by exploiting additional structure. An example of interest is the family of monotone distributions. By a result of Birge [4] (see also [14] for an adaptation to the discrete case), every monotone distribution over $[n]$ is $\varepsilon$-close in $\ell_1$-distance to a $k$-histogram distribution, for $k = O(\varepsilon^{-1}\log n)$. Hence, an application of the above theorem yields a distributed estimation algorithm for monotone distributions. The main insight here is that each monotone distribution is well-approximated by an oblivious histogram, i.e., one whose intervals are the same for each monotone distribution. This allows us to essentially reduce the learning problem to that of learning a discrete distribution over the corresponding domain size. A reduction in the opposite direction yields the matching lower bound. Please refer to the full version for more details.

## 1.3 Comparison to Related Work

Recent works [40, 21, 24, 5] study the communication cost of mean estimation problems of structured, parametrized distributions. These works develop powerful information theoretic tools to obtain lower

bounds for parameter estimation problems. In what follows, we briefly comment why we need to develop new techniques by pointing out fundamental differences between the two problems.

First, our most general results on distributed density estimation do not assume any structure on the distribution (and thus, our learning algorithms are agnostic). This is in contrast to the problems considered before, where the concept classes are restricted (Gaussians, linear separators) and enjoy a lot of structure, which is often leveraged during the design of estimators.

Secondly, while we also consider more structured distributions (monotone, $k$-histograms), the techniques developed in the study of distributed parameter estimation do not apply to our problems. Specifically, those results reduce to the problem of learning a high-dimensional vector (say, where each coordinate parametrizes a spherical Gaussian distribution), where the value at each coordinate is *independent* of the others. The results in distributed parameter estimation crucially use the coordinate independence feature. The lower bounds essentially state that the communication cost of a $d$-dimensional parameter vector with independent components grows proportionally to the dimension $d$, and hence one needs to estimate each coordinate separately.

## 2 Preliminaries

**Notation.** For any positive integer $n$, we write $[n]$ to denote $\{1, \ldots, n\}$, the set of integers between 1 and $n$. We think of a probability distribution $P$ on $[n]$ as a vector of probabilities $(p_1, \ldots, p_n)$ that sum up to 1. We write $X \sim P$ to denote that a random variable $X$ is drawn from $P$. Sometimes we use the notation $P(i)$ to denote $\mathbb{P}[X = i]$, where $X \sim P$. We consider three families of discrete distributions:

- $\mathcal{D}_n$: the family of unstructured discrete distributions on $[n]$,
- $\mathcal{H}_{n,k}$: the family of $k$-histogram distributions on $[n]$,
- $\mathcal{M}_n$: the family of monotone distributions on $[n]$.

We use $\ell_p$ metrics on spaces of probability distributions. For two distributions $P$ and $P'$ on $[n]$, their $\ell_p$-distance, where $p \in [1, \infty)$, is defined as

$$\|P - P'\|_p := \left( \sum_{i=1}^n |P(i) - P'(i)|^p \right)^{1/p}.$$

In this work we focus on the cases of $p = 1$ and $p = 2$, in which $\|P - P'\|_1 = \sum_{i=1}^n |P(i) - P'(i)|$ and $\|P - P'\|_2 = \sqrt{\sum_{i=1}^n (P(i) - P'(i))^2}$.

For a given distribution $Q \in \mathcal{D}_n$ and family $\mathcal{P} \subseteq \mathcal{D}_n$ of distributions, we denote the $\ell_p$-distance of $Q$ to $\mathcal{P}$ as $\mathrm{dist}_p(Q, \mathcal{P}) := \inf_{P \in \mathcal{P}} \|Q - P\|_p$.

**Packings and the Packing Number.** Let $(X, \|\cdot\|_p)$ be a normed space, $E \subset X$, and $r > 0$ be a radius. $E' = \{e_1, \ldots, e_n\} \subset E$ is an $(r, p)$-packing of $E$ if $\min_{i \neq j} \|e_i - e_j\|_p > r$. The $(r, p)$-packing number $N_r^{\mathrm{pack}}(E, p)$ is the cardinality of the largest $(r, p)$-packing of $E$, i.e., $N_r^{\mathrm{pack}}(E, p) := \sup\{|E'| \mid E' \subset E \text{ is an } (r, p)\text{-packing of E}\}$.

**Density Estimation.** We now formally introduce density estimation problems considered in this paper. First, for a given $n \in \mathbb{Z}_+$, let $\mathcal{P} \subseteq \mathcal{D}_n$ be a family of distributions on $[n]$, $\varepsilon \in [0, \infty)$, and $p \in [1, \infty)$. The goal of the *density estimation problem* $\mathrm{DE}(\mathcal{P}, p, \varepsilon)$ is to output, for any unknown distribution $P \in \mathcal{P}$, a distribution $Q \in \mathcal{D}_n$ such that $\|P - Q\|_p \leq \varepsilon$. Note that in this problem, we are guaranteed that the unknown distribution belongs to $\mathcal{P}$.

Now we define a version of the problem that allows inputs from outside of the class of interest. For a given $n \in \mathbb{Z}_+$, let $\mathcal{P} \subseteq \mathcal{P}$ be a family of distributions on $[n]$. Also let $\varepsilon \in [0, \infty)$, $p \in [1, \infty)$, and $\alpha \in [1, \infty)$. The goal of the *agnostic density estimation problem* $\mathrm{ADE}(\mathcal{P}, p, \varepsilon, \alpha)$ is to output, for *any* unknown distribution $P \in \mathcal{D}_n$, a distribution $Q \in \mathcal{D}_n$ such that $\|P - Q\|_p \leq \alpha \cdot \mathrm{dist}_p(P, \mathcal{P}) + \varepsilon$, with high probability. The reason for this version of the problem is that in practice one often has to deal with noisy or non-ideal data. Hence if the unknown distribution is close to belonging to a class $\mathcal{P}$, we wish to output a near distribution as well.

**Estimators and Sample Complexity.** For any distribution estimation problem $A$ involving an unknown distribution $P$—such as $\mathrm{DE}(\mathcal{P}, p, \varepsilon)$ and $\mathrm{ADE}(\mathcal{P}, p, \varepsilon, \alpha)$ defined above—we now introduce the notion of an estimator. For any $m \in \mathbb{N}$, an *estimator* $\theta : [n]^m \times \{0,1\}^\infty \to \mathcal{D}_n$ is a function that takes a sequence $\vec{\mathrm{X}} = (X_1, \ldots X_m)$ of $m$ independent samples from $P$ and sequence $R$ of uniformly and independently distributed random bits, and outputs a hypothesis distribution $\widehat{P} := \theta(\vec{\mathrm{X}}, R)$. We say that the estimator *solves $A$ with probability* $1 - \delta$ if for any unknown distribution $P$ allowed by the formulation of problem $A$, the probability that $\widehat{P}$ is a correct solution to $A$ is at least $1 - \delta$. For instance, if $A$ is the $\mathrm{ADE}(\mathcal{P}, p, \varepsilon, \alpha)$ problem, the hypothesis distribution $\widehat{P}$ produced by the estimator should satisfy the following inequality for any distribution $P \in \mathcal{D}_n$:

$$\mathbb{P}\left[ \|\widehat{P} - P\|_p \leq \alpha \cdot \mathrm{dist}_p(P, \mathcal{P}) + \varepsilon \right] \geq 1 - \delta.$$

The *sample complexity of $A$ with error $\delta$*, which we denote $\mathcal{SC}_\delta(A)$, is the minimum number of samples $m$, for which there exists an estimator $\theta : [n]^m \times \{0,1\}^\infty \to \mathcal{D}_n$ that solves $A$ with probability $1 - \delta$.

As a simple application of this notation, note that $\mathcal{SC}_\delta(\mathrm{DE}(\mathcal{P}, p, \varepsilon)) \leq \mathcal{SC}_\delta(\mathrm{ADE}(\mathcal{P}, p, \varepsilon, \alpha))$ for any $\alpha \in [1, \infty)$. This follows from the fact that in $\mathrm{DE}(\mathcal{P}, p, \varepsilon)$, one has to solve exactly the same problem but only for a subset of input distributions in $\mathrm{ADE}(\mathcal{P}, p, \varepsilon, \alpha)$. Since the input $P$ for $\mathrm{DE}(\mathcal{P}, p, \varepsilon))$ comes from $\mathcal{P}$, we have $\mathrm{dist}_p(P, \mathcal{P}) = 0$.

**Communication Complexity of Density Estimation.** In all of our communication models, when a player wants to send a message, the set of possible messages is *prefix-free*, i.e., after fixing both the randomness and the set of previous messages known to the player, there are no two possible messages such that one is a proper prefix of the other. Furthermore, for a protocol $\Pi$ in any of them, we write $\mathrm{Cost}_\mathcal{P}(\Pi)$ to denote the *(worst-case) communication cost of $\Pi$ on $\mathcal{P}$* defined as the maximum length of messages that can be generated in the protocol if the unknown distribution belongs to $\mathcal{P}$. Similarly, we write $\overline{\mathrm{Cost}}_\mathcal{P}(\Pi)$ to denote the *expected communication cost of $\Pi$ on $\mathcal{P}$* defined as the maximum expected total length of messages exchanged, where the maximum is taken over all unknown distributions in $\mathcal{P}$ and the expectation is taken over all assignments of samples to machines and settings of public randomness. The following inequality always holds: $\overline{\mathrm{Cost}}_\mathcal{P}(\Pi) \leq \mathrm{Cost}_\mathcal{P}(\Pi)$.

> **Simultaneous communication.** In the *simultaneous* communication model, each sample-holding player sends a message to the referee once, based only on the samples she holds and public randomness.
>
> For a density estimation problem $A$, let $\mathcal{P}$ be the family of possible unknown distributions $P$. We write $\mathcal{CC}_{s,\delta}^{\rightarrow}(A)$ to denote $(s, \delta)$-*simultaneous communication complexity of $A$* defined as the minimum $\mathrm{Cost}_\mathcal{P}(\Pi)$ over all simultaneous communication protocols $\Pi$ that solve $A$ with probability at least $1 - \delta$ for any $P \in \mathcal{P}$ with $s$ samples per sample-holding player and an arbitrary number of sample-holding players.
>
> **Blackboard communication.** In this model, each message sent by each player is visible to all players. The next player speaking is uniquely determined by the previously exchanged messages and public randomness. We use this model to prove lower bounds. Any lower bound in this model applies to the previous communication models. More specifically, we show lower bounds for the *average* communication complexity, which we define next.
>
> For a density estimation problem $A$, let $\mathcal{P}$ be the family of possible unknown distributions $P$. We write $\overline{\mathcal{CC}}_{s,\delta}(A)$ to denote $(s, \delta)$-*average communication complexity of $A$* defined as the infimum $\overline{\mathrm{Cost}}_\mathcal{P}(\Pi)$ over all blackboard protocols $\Pi$ that solve $A$ with probability at least $1 - \delta$ for any $P \in \mathcal{P}$ with $s$ samples per sample-holding player and an arbitrary number of sample-holding players.

The communication complexity notions that we just introduced remain in the following relationship.

**Claim 1.** *For any density estimation problem $A$,*

$$\overline{\mathcal{CC}}_{s,\delta}(A) \leq \mathcal{CC}_{s,\delta}^{\rightarrow}(A).$$

The claim follows from the fact that simultaneous communication is a specific case of blackboard communication. Additionally, expected communication cost lower bounds worst-case communication

cost. All lower bounds that we prove are on the average communication complexity in blackboard communication.

**A Trivial Upper Bound.** There is always a trivial protocol that leverages the sample complexity of the density estimation. Since $\mathcal{SC}_\delta(A)$ samples are enough to solve the problem, it suffices that sample-holding players communicate this number of samples to the referee. Since each sample can be communicated with at most $\lceil \log n \rceil$ bits, we obtain the following upper bound on the simultaneous communication complexity.

**Claim 2.** *For any density estimation problem $A$ and any $s \geq 1$,*
$$\mathcal{CC}^{\rightarrow}_{s,\delta}(A) \leq \mathcal{SC}_\delta(A) \cdot \lceil \log n \rceil .$$

In this paper, we investigate whether there exist protocols that significantly improve on this direct upper bound.

**Randomness.** All our protocols are deterministic (more precisely, depend only the randomness coming from samples provided by the samples from the hidden distribution). On the other hand our lower bounds apply to all protocols, also those using an arbitrary amount of public randomness (i.e., pre-shared randomness).

# 3 Our Techniques

In this section, we provide a high-level description of the main ideas in our upper and lower bounds. We defer the details of upper and lower bounds for monotone distributions to the full version of the paper.

## 3.1 Overview of Algorithmic Ideas

We start by describing the main ideas in our distributed learning algorithms.

**Robustly Learning Histograms in $\ell_1$-Distance.** We will require the following definition:

**Definition 1.** *(Distribution flattening) Let $P$ be a distribution over $[n]$ and let $\mathcal{I} = \{I_i\}_{i=1}^\ell$ be a partition of $[n]$ into disjoint intervals. We denote by $\bar{P}_\mathcal{I}$ the distribution over $[n]$, where*
$$\bar{P}_\mathcal{I}(i) = \frac{\sum_{k \in I_j} P(k)}{|I_j|}, \qquad \forall j \in [\ell], i \in I_j .$$

*This means that $\bar{P}_\mathcal{I}$ is obtained by spreading the total mass of an interval uniformly in the interval.*

Our upper bounds in this setting crucially depend on the following norm from Vapnik-Chervonenkis (VC) theory [39], known as the $\mathcal{A}_k$ norm (see, e.g., [18]).

**Definition 2** ($\mathcal{A}_k$ norm). *For any function $f : [n] \to \mathbb{R}$, we define the $\mathcal{A}_k$ norm of $f$ as*
$$\|f\|_{\mathcal{A}_k} = \sup_{I_1,\ldots,I_k} \sum_{i=1}^{k} |f(I_i)| ,$$

*where for any set $S \subseteq [n]$, we let $f(S) = \sum_{i \in S} f(i)$ and the supremum is taken over disjoint intervals.*

In other words, the $\mathcal{A}_k$ norm of $f$ is the maximum norm of any flattening of $f$ into $k$ interval pieces.

Our distributed algorithms crucially rely on the following building blocks:

**Theorem 4** ([2]). *Let $P : [n] \to \mathbb{R}$ be a distribution, and let $\widehat{P} : [n] \to \mathbb{R}$ be a distribution such that $\|P - \widehat{P}\|_{\mathcal{A}_k} \leq \varepsilon$. There is an efficient algorithm* LEARNHIST$(\widehat{P}, k, \varepsilon)$ *that given $\widehat{P}$, outputs a $k$-histogram $h$ such that $\|P - h\|_1 \leq 3\mathrm{OPT}_k + O(\varepsilon)$, where $\mathrm{OPT}_k = \min_{h \in \mathcal{H}_{n,k}} \|P - h\|_1$.*

This theorem says that if we know a proxy to $P$ that is close in $\mathcal{A}_k$-norm to $P$, then this gives us enough information to construct the best $k$-histogram fit to $P$. Moreover, this is the *only* information we need to reconstruct a good $k$-histogram fit to $P$. The following well-known version of the VC-inequality states that the empirical distribution after $O(k/\varepsilon^2)$ samples is close to the true distribution in $\mathcal{A}_k$-norm:

**Theorem 5** (VC inequality, e.g., [18]). *Fix $\varepsilon, \delta > 0$. Let $P : [n] \to \mathbb{R}$ be a distribution, and let $Q$ be the empirical distribution after $O(\frac{k + \log 1/\delta}{\varepsilon^2})$ samples from $P$. Then with probability at least $1 - \delta$, we have that $\|P - Q\|_{\mathcal{A}_k} \leq \varepsilon$.*

These two theorems together imply (via the triangle inequality) that in order to learn $P$, it suffices to construct some distribution $\widehat{P}$ such that the empirical distribution $Q$ is close to $\widehat{P}$ in $\mathcal{A}_k$-norm. After we construct this $\widehat{P}$, we can run LEARNHIST at a centralized server, and simply output the resulting hypothesis distribution. Thus, the crux of our distributed algorithm is a communication-efficient way of constructing such a $\widehat{P}$.

We achieve this as follows. First, we learn a partition $\mathcal{I}$ of $[n]$ such that on each interval $I \in \mathcal{I}$, either $|I| = 1$ and $Q(I) \geq \Omega(\varepsilon/k)$, or we have $Q(I) \leq O(\varepsilon/k)$. We then show that if we let $\widehat{P}$ be the flattening of $Q$ over this partition, then $\widehat{P}$ is $\varepsilon$-close to $P$ in $\mathcal{A}_k$-norm. To find this partition, we repeatedly perform binary search over the the domain to find intervals of maximal length, starting at some fixed left endpoint $\ell$, such that the mass of $Q$ over that interval is at most $O(\varepsilon/k)$. We show that the intervals in $\mathcal{I}$ can be found iteratively, using $O(m \log ms \log n)$ bits of communication each, and that there are at most $O(k/\varepsilon)$ intervals in $\mathcal{I}$. This in turn implies a total upper bound of $\tilde{O}(mk \log n/\varepsilon)$ bits of communication, as claimed.

We also show a black-box reduction for robustly learning $k$-histograms. It improves on the communication cost when the domain size is very large. Specifically, we show:

**Lemma 1.** *Fix $n \in \mathbb{N}$, and $\varepsilon, \delta > 0$. Suppose for all $1 \leq n' \leq n$, there is a robust learning algorithm for $\mathcal{H}_{n',k}$ with $s$ samples per server and $m$ servers, using $B(k, n,' m, s, \varepsilon)$ bits of communication, where $ms \geq \Omega((k + \log 1/\delta)/\varepsilon^2)$. Then there is an algorithm which solves $\mathcal{H}_{n,k}$ using $O(B(k, O(k/\varepsilon), s, \varepsilon) + \frac{k}{\varepsilon} \log n)$ bits of communication.*

In other words, by increasing the communication by an additive factor of $\frac{k}{\varepsilon} \log n$, we can replace the domain size $n$ with $O(k/\varepsilon)$. This is crucial for getting tighter bounds in certain regimes.

**Learning Histograms in $\ell_2$-Distance.**   We now describe our algorithm for learning $k$-histograms in $\ell_2$. We first require the following folklore statistical bound:

**Lemma 2** (see e.g. [1]). *Fix $\varepsilon, \delta > 0$ and a distribution $P : [n] \to \mathbb{R}$. Let $Q$ be the empirical distribution with $O(\log(1/\delta)/\varepsilon)$ i.i.d. samples from $P$. Then with probability $1 - \delta$, we have $\|P - Q\|_2^2 \leq \varepsilon$.*

This lemma states that it suffices to approximate the empirical distribution $Q$ in $\ell_2$ norm. We now describe how to do so.

Our first key primitive is that using the celebrated Johnson-Lindenstrauss lemma [28], it is possible to get an accurate estimate of $\|x\|_2^2$ when server $i$ has access to $x_i$ and $x = \sum x_i$, where each server communicates at most logarithmically many bits, regardless of the dimension of $x$. Moreover, we can do this for $\text{poly}(n)$ many different $x$'s, even without shared randomness, by communicating only $O(\log n \log \log n)$ bits once at the beginning of the algorithm and constantly many bits per call afterwards. In particular, we use this to approximate

$$e_I = \sum_{i \in I} (Q(i) - Q(I))^2 \ ,$$

for all intervals $I \subseteq [n]$.

Perhaps surprisingly, we are now able to give an algorithm that outputs the best $O(k \log n)$-histogram approximation to $Q$ in $\ell_2$, which only accesses the distribution via the $e_I$. Moreover, we show that this algorithm needs to query only $O(k \log n)$ such $e_I$. Since each query to $e_I$ can be done with logarithmically many bits per server, this yields the claimed communication bound of $\tilde{O}(mk \log n)$. Roughly speaking, our algorithm proceeds as follows. At each step, it maintains a partition of $[n]$. Initially, this is the trivial partition containing just one element: $[n]$. Then in every iteration it finds the $2k$ intervals in its current partition with largest $e_I$, and splits them in half (or splits them all in half if there are less than $2k$ intervals). It then repeats this process for $\log n$ iterations, and returns the flattening over the set of intervals returned. By being somewhat careful with how we track error, we are able to show that this in fact only ever requires $O(k \log n)$ queries to $e_I$. While this algorithm is quite simple, proving correctness requires some work and we defer it to the full version.

### 3.2 Proof Ideas for the Lower Bounds

We now give an overview of proofs of our lower bounds.

**Interactive Learning of Unstructured Distributions.** We start with the most sophisticated of our lower bounds: a lower bound for unstructured distributions with one sample per player and arbitrary communication in the blackboard model. We show that $\Omega((n/\varepsilon^2) \log n)$ bits of communication are needed. Thid is optimal and implies that in this case, there is no non-trivial protocol that saves more than a constant factor over the trivial one (in which $O(n/\varepsilon^2)$ samples are fully transmitted). In order to prove the lower bound, we apply the information complexity toolkit. Our lower bound holds for a family of nearly uniform distributions on $[n]$, in which each pair of consecutive elements, $(2i-1, 2i)$, have slightly perturbed probabilities. In the uniform distribution each element has probability $1/n$. Here for each pair of elements $2i-1$ and $2i$, we set the probabilities to be $\frac{1}{n}(1 + 100\delta_i\varepsilon)$ and $\frac{1}{n}(1 - 100\delta_i\varepsilon)$, where each $\delta_i$ is independently selected from the uniform distribution on $\{-1, 1\}$. Each such pair can be interpreted as a single slightly biased coin. We show that the output of any good learning protocol can be used to learn the bias $\delta_i$ of most of the pairs. This implies that messages exchanged in any protocol that is likely to learn the distribution have to reveal most of the biases with high constant probability.

Intuitively, the goal in our analysis is to show that if a player sends much fewer than $\log n$ bits overall, this is unlikely to provide much information about that player's sample and help much with predicting $\delta_i$'s. This is done by bounding the mutual information between the transcript and the $\delta_i$'s. It should be noted that our lower bound holds in the interactive setting. That is, players are unlikely to gain much by *adaptively* selecting when to continue providing more information about their samples. The details of the proof are deferred to the full version.

**Packing Lower Bounds.** Some of our lower bounds are obtained via the construction of a suitable packing set. We use the well-known result that the logarithm of the size of the packing set is a lower bound on the communication complexity. This follows from using the well-known reduction from estimation to testing, in conjunction with Fano's inequality.

## 4 Conclusion and Open Problems

This work provides the first rigorous study of the communication complexity of nonparametric distribution estimation. We have obtained both negative results (tight lower bounds in certain regimes) and the first non-trivial upper bounds for a range of structured distributions.

A number of interesting directions remain. We outline a few of them here:

1. The positive results of this work focused on discrete univariate structured distributions (e.g., histograms and monotone distributions). For what other families of structured distributions can one obtain communication-efficient algorithms? Studying *multivariate* structured distributions in this setting is an interesting direction for future work.

2. The results of this paper do not immediately extend to the continuous setting. Can we obtain positive results for structured *continuous* distributions?

3. It would be interesting to study related inference tasks in the distributed setting, including hypothesis testing and distribution property estimation.

**Acknowledgments**

The authors would like to thank the reviewers for their insightful and constructive comments. ID was supported by NSF Award CCF-1652862 (CAREER) and a Sloan Research Fellowship. EG was supported by NSF Award CCF-1649515. JL was supported by NSF CAREER Award CCF-1453261, CCF-1565235, a Google Faculty Research Award, and an NSF Graduate Research Fellowship. AN was supported in part by a grant from the Purdue Research Foundation and NSF Awards CCF-1618981 and CCF-1649515. LS was funded by a Google PhD Fellowship.

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
