[Reviews · NeurIPS 2017]

Reviewer 1



This paper studies the problem of learning discrete distributions in the distributed setting, where random samples from an unknown discrete distribution are evenly distributed over machines, and the machines can communicate with a referee that at the end needs to output an estimation of the distribution. This problem is important across various applications and also theoretically interesting, but has been studied systematically. This paper provides a set of results that cover the cases of estimating unstructured and structured distributions in l1 and l2 distances. For unstructured distributions, if each machine has only one sample, then the trivial communication protocal of sending all the data to the referee has optimal communication up to constant factors. Lower and upper bounds are also provided when the machines have more than one sample though with a gap. For structured distributions, the paper studies k-histogram and monotone distributions. In different regimes about the number of samples on each machine, protocols are proposed that achieve better communication guarantees than the trivial ones. In some regimes the bounds are tight. It is impressive that the study is quite complete, covering almost all regimes (except the lower bounds for learning of monotone distributions in l1). The tables in the supplementary materials are informative and I suggest moving some of them to the main text. The presentation is clear and related works are properly cited. Overall, the paper proposes a direction that is both empirically important and theoretically interesting, and provides many highly non-trival results, and insightful conceptual messages. There are a lot of interesting open questions, like improving the bounds in the paper, considering other important structured distributions, considering other metrics besides l1 and l2, etc. -- Line 349: learnking -> learning

Reviewer 2



Summary; The paper studies the classical problem of estimating the probability mass function (pmf) of a discrete random variable given iid samples, but distributed among different nodes. The key quantity of interest is how much communication must be expended by each node (in a broadcast, but perhaps interactive, setting) to a central observer which then outputs the estimate of the underlying pmf. The main results of the paper areclearly stated and are easy to follow. The results mostly point out that in the worst case (i.e., no assumptions on the underlying pmf) there is nothing better for each node to do than to communicate its sample to the central observer. The paper addresses a central topic of a long line of recent works on distributed parameter estimation. The distinguishing feature here is to have as few parametric assumptions on the underlying distribution as possible. 1. In the case when no assumption at all is made on the underlying distribution (other than the alphabet size), the main result is to show that essentially nothing better can be achieved than when all nodes communicate their raw samples (noninteractively) to the central observer. The upper bound is hence trivial and doesnt take much to analyze. The key innovation is in the lower bound where the authors consider a specific near-uniform distribution (where neighboring letters have probabilities that are parametrically (\delta_i in the notation of the paper) close to each other). Then the authors show that any (interactive) protocol allows the central observer to learn these \delta_i parameters and which itself is tantamount to learning the sample itself. The key calculation is to bound the mutual information between the transcripts and the \delta_i parameters which is then fed into a standard hypothesis testing framework (Fano's inequality). Although this machinery is standard, I found the setup of the hypothesis test creative and interesting and novel. Overall, this part of the paper is of fundamental interest (no assumptions at all on the underlying hypothesis) and nicely carried out. 2. I am less sure about the two `structured' settings considered in this paper: k-histograms and monotone pmfs. I can imagine monotone pmfs coming up in some application domain, but far less sure about why k-histograms makes sense. The main challenging aspect of this setting is that only k intervals are allowed, but their position is arbitrary. I understand the formulation makes for pretty math (and fits in nicely with a bunch of existing literature), but cannot see any canonical/practical setting where allowing this flexibility makes sense. I would appreciate it if the authors can elaborate on this point in the rebuttal stage. 3. In the context of the last sentence of the previous comment, I would also like to see some discussion by authors on any practical implications of this work. For instance the authors mention that the problem is very well motivated and cite some works from the literature [44, 40, 30, 54, 47]. Of these, I would guess that [47] is the only recent practical work (the others are either more than a decade old or also correspond to an already stylized setting) -- on this topic, can the authors please fix the reference [40] so that it includes its title? I would like to see concrete settings where the structured settings are motivated properly and the authors actually try out their algorithms (along with competing baselines) in that setting. In the context of this paper, I would like to see a (brief) discussion of how the settings might be related to the distributed computing revolution underlying modern data centers. Most NIPS papers have a section where they discuss their algorithms in the context of some concrete practical setting and report empirical results. 3. The schemes proposed in the two structured settings are more sophisticated than the baseline one (where the nodes simply communicate their samples directly), but are directly motivated from their centralized counterparts. In particular, the k-histogram setting uses ideas from [3] and approximation in the \A_k norm and the monotone setting from the classical work of Birge. Overall, the proofs are correct and represent a reasonable innovation over a long line of work on this topic. Summary: I recommend this submission as far better suited to COLT or even SODA.

Reviewer 3



This paper considers the problem of learning a discrete distribution in the distributed model. The samples from the distribution is partitioned into a number of machines, and the goal is to efficiently communicate to a central server that will learn the distribution. They show that if the distribution is unstructured, the naive protocol (of each machine just sending over its entire sample) is the best one can do. On the other hand, the authors consider certain families of structured distributions where it is possible to non-trivially improve upon this naive strategy. The paper is very well written. In particular, I commend the authors' simplified theorem statements in the main text, while leaving more precise versions to the appendices. The proof sketches in the main text are illuminating and sufficient to understand the ideas. There are significant gaps in the upper and lower bounds in Theorems 2.2, 2.3, and 2.4, especially in terms of epsilon. The authors should explain this, preferably right after the theorem statements. Its not clear to me if these are inevitable or if they are simply artifacts of the analysis. That said, this is a solid submission, and I recommend its acceptance. Minor: - The sentence starting in Line 139 is incomplete. - Line 349: learnking-->learning